# Urban Expansion Monitoring Based on the Digital Surface Model—A Case Study of the Beijing–Tianjin–Hebei Plain

**Yanping Wang** [1,*], **Pinliang Dong** [2], **Shunbao Liao** [1] , **Yueqin Zhu** [3], **Da Zhang** [1] **and Na Yin** [1]

1 Department of Ecology and Environment, Institute of Disaster Prevention, Sanhe 065201, China; liaoshunbao@cidp.edu.cn (S.L.); zhangda@cidp.edu.cn (D.Z.); yinna@cidp.edu.cn (N.Y.)
2 Department of Geography and the Environment, University of North Texas, Denton, TX 76203, USA; pinliang.dong@unt.edu
3 National Institute of Natural Hazards, Ministry of Emergency Management of the People's Republic of China, Haidian, Beijing 100085, China; yueqinzhu@163.com
* Correspondence: wangyanping@cidp.edu.cn

**Abstract:** Although urban expansion statistics have been widely carried out, large-scale and rapid monitoring is still worth doing in order to improve the efficiency of statistics, as well as make up for the omissions and deficiencies of construction expansion statistics with multi-year intervals. This paper presents a study of urban expansion in the Beijing–Tianjin–Hebei plain based on ALOS Global Digital Surface Model "ALOS World 3D-30 m" (AW3D30 DSM), Shuttle Radar Topography Mission (SRTM) DSM, and Landsat 7 ETM+ images. Through the evaluation of errors and the elimination of non-building changes, a relatively objective result is derived. The neighborhood block statistics of the construction height expansion reveal that from 2000 to 2009, the largest centralized construction expansion mainly occurred between the Second Ring Road and the Fifth Ring Road of Beijing, followed by Yizhuang, Shunyi, Tianjin Central City, and Langfang. Zonal statistics also show a significant imbalance in the expansion of construction in the counties of the Beijing–Tianjin–Hebei plain. For example, Chaoyang, Dongcheng, Xicheng, Xuanwu, Chongwen, Nankai, Heping, and Hexi have a larger construction expansion; however, other counties present a relatively slow rate of building expansion. Furthermore, the correlation coefficient between the statistical average building height expansion per unit area (ABHE, by our method) and the actual average completed building floor space per unit area (ACBFS) derived from the Beijing Statistical Yearbook (BSY) is 0.9436, which proves that this method is feasible. With the continuous improvement of DSM data quality in the future, the method proposed in this paper can provide rapid and large-scale statistics to study more urban construction expansion in the world.

**Keywords:** three-dimensional urban expansion; building height; AW3D30 DSM; SRTM DSM; Beijing–Tianjin–Hebei plain

## 1. Introduction

Urbanization is an inevitable trend of global social and economic development. The expansion speed of urban construction is of great significance to studying the urban growth model [1–3], as well as identifying urbanization processes [4]. Arbitrary expansion of urban land use has brought great challenges to groundwater exploitation, ground bearing capacity [5,6], air quality [7], and even leads to many harmful chain reactions [8–11]. The authorities need reliable and rapid statistics on the three-dimensional expansion or expansion velocity of buildings in the jurisdiction in order to plan urban development and formulate development policy [12,13]. However, it is difficult to achieve rapid and large-scale statistics due to a large amount of construction volume and intricate demolition information [14,15].

Remote sensing technology provides a convenient solution for statistics and comparison in this task. At present, many methods and cases have been presented in the literature.

For example, urban land-use change detection based on remote sensing images has been used in many case studies [16–23]. In recent years, night-time light data are also used to study urban expansion [24]. This method is very effective for the horizontal expansion of buildings, but it is not sensitive to the vertical expansion of the city.

Megacities usually begin as horizontal expansion initially, and then grow vertically in the already developed areas due to the shortage of land resources [25]. Therefore, capturing the height change of urban buildings in time is an effective way to dynamically monitor the urban development process. Nowadays, building height can be retrieved from synthetic aperture radar (SAR) imagery [26,27]. Furthermore, high-precision building height information can be extracted from photon-counting LiDAR data [28,29]. Other predecessors focus on generating a digital surface model from optical stereo image pairs or SAR images, and then compare the height differences of different DSMs to identify newly constructed buildings, as well as demolitions in dense residential areas [30–34]. Nevertheless, more details, such as the exclusion of non-construction factors [35–38], are still worth further study. Moreover, openly accessible information at high spatial resolution is still missing for complete countries or regions [39].

Multi-temporal DSMs released for free worldwide, including those already opened and those to be opened in the future, have the advantage of rapidly detecting building height changes in large areas, and even on a global scale. However, due to many uncertain factors such as data errors and mixed non-building changes, they are less used for building expansion statistics [40].

Our research mainly focuses on using AW3D30 DSM and SRTM DSM as an example to calculate building height expansion. By registering DSMS data, verifying the change of sample building height, and excluding the non-building height shift, the three-dimensional expansion of the building is calculated. The result can be converted to the average completed building floor space per unit area (ACBFS) to compensate for statistical deficiencies.

## 2. Study Area and Data

### 2.1. Study Area

As shown in Figure 1, the study area is mainly located in the Beijing–Tianjin–Hebei plain. It includes several major counties in the Beijing–Tianjin–Hebei region. This area is primarily a plain area with a topographic slope of less than 5 degrees, which is the urban agglomeration in northern China. There are many buildings distributed, and the buildings expand rapidly and unevenly. During the study period, the total population of the study area was promoted from 18.2 million in 2001 to 20.7 million in 2009. Now, it has reached 40 million in 2022. Compared with other regions, this area has a relatively high urbanization rate, from 43.6% in 2001 to 52.7% in 2009 [41–50]. According to Hai-quan YANG's research, the overall efficiency of urban land use in Beijing–Tianjin–Hebei was high, and had a relatively small redundancy. However, it had a deceleration of 2.86% from 2000 to 2009 [51].

### 2.2. Data

The data used in the study include SRTM DSM and AW3D30 DSM. The Shuttle Radar Topography Mission (SRTM) is an international research effort that obtained digital elevation models on a near-global scale from 56° S to 60° N during the 11-day STS-99 mission in February 2000. SRTM DEM data was released by NASA in 2003, with a resolution of one arcsec, an EGM96 (gravity model 1996) vertical datum, and a WGS84 horizontal datum, which is generated by interferometric synthetic aperture radar (InSAR) technology [52–57]. AW3D30 is a global digital surface model (DSM) with a horizontal resolution of approximately 30 m (basically one arcsec) created by the Panchromatic Remote-Sensing Instrument for Stereo Mapping (PRISM), which was an optical sensor onboard the Advanced Land Observing Satellite "ALOS". AW3D30 DSM uses 3 million scene archives acquired by the PRISM panchromatic optical sensor on the Advanced Land Observing Satellite "DAICHI" (ALOS), which operated from 2006 to 2011. In this study area, by comparing the height

of the built-up area with the building construction time, the archived 2009 images were used in this study. AW3D30 DSM is provided in height above sea level vertical datum and WGS84 ellipsoidal horizontal datum [58,59]. The comparison of the two DSM data parameters is shown in Table 1.

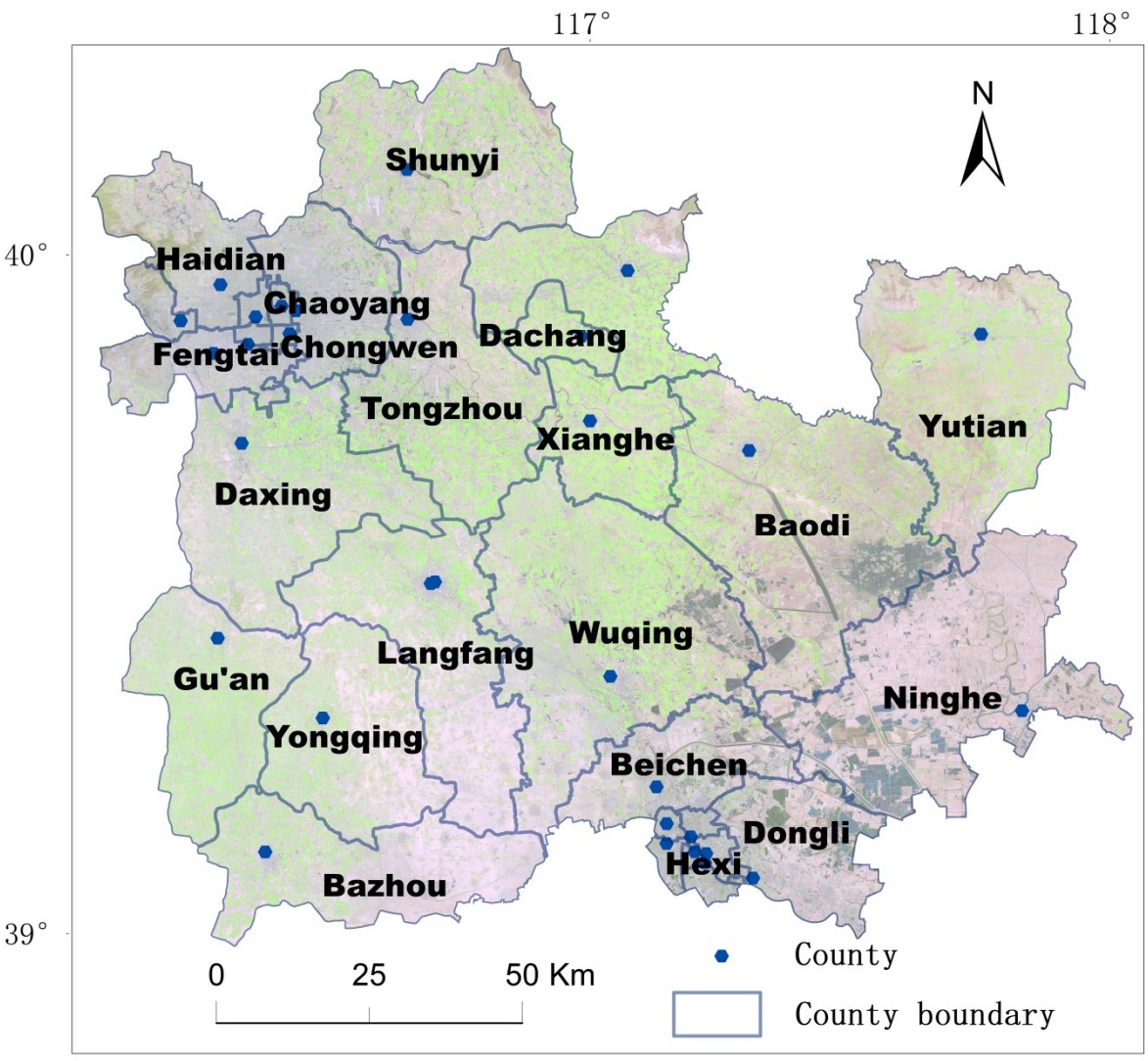

**Figure 1.** Location of the study area.

**Table 1.** Parameter of SRTM DSM and AW3D30 DSM.

|  | SRTM DSM | AW3D30 DSM |
|---|---|---|
| Name | Shuttle Radar Topography Mission (SRTM) | ALOS World 3D-30 m (AWD3D30) |
| Imaging time | 2 November 2000 | 2006–2011 |
| Sensor | JPL | PRISM |
| Resolution | 30 m | 30 m |
| Plane datum | WGS84 | WGS84 |
| Elevation datum | EGM96 (Earth Gravitational Model 1996) | Height above sea level |
| Absolute Geolocation Error | 8 m (Root Mean Square Error in Eurasia) | 5 m (Root Mean Square Error) |
| Absolute Height Error | 6 m (Root Mean Square Error in Eurasia) | 4 m (Root Mean Square Error) |
| Relative Height Error | 5 m (Root Mean Square Error in the study area) | 4 m (Root Mean Square Error) |

Landsat 7 ETM+ images consist of eight spectral bands with a spatial resolution of 30 m for bands 1 to 7 [60]. The panchromatic band 8 has a resolution of 15 m (Table 2). The approximate scene size is 170 km north-south by 183 km east-west (106 mi by 114 mi).

**Table 2.** Landsat 7 Enhanced Thematic Mapper Plus (ETM+).

| Bands | Wavelength (Micrometers) | Resolution (Meters) |
|---|---|---|
| Band 1—Blue | 0.45–0.52 | 30 |
| Band 2—Green | 0.52–0.60 | 30 |
| Band 3—Red | 0.63–0.69 | 30 |
| Band 4—Near Infrared (NIR) | 0.77–0.90 | 30 |
| Band 5—Shortwave Infrared (SWIR) 1 | 1.55–1.75 | 30 |
| Band 6—Thermal | 10.40–12.50 | 60 (30) |
| Band 7—Shortwave Infrared (SWIR) 2 | 2.09–2.35 | 30 |
| Band 8—Panchromatic | 0.52–0.90 | 15 |

## 3. Methodology

The methodology includes the following major steps (Figure 2): (1) the registration of the accuracy verification of the two DSM data, (2) the detection of elevation change information, (3) the verification and evaluation of elevation change accuracy, (4) the exclusion of non-building changes, and (5) the calculation of building expansion or demolition amount. The main software used for data processing are ArcGIS and ENVI. More details of the steps are described below.

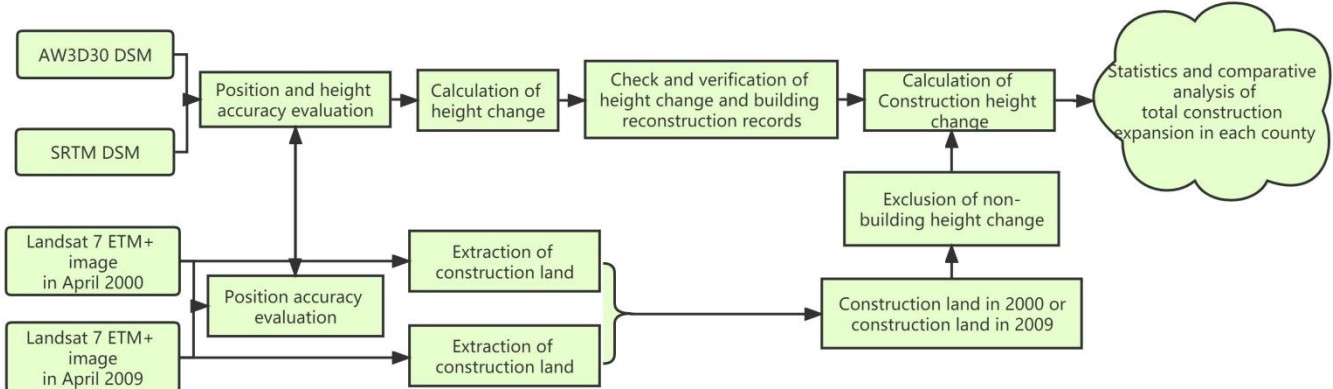

**Figure 2.** Methodology flowchart.

### 3.1. Registration Accuracy Verification of the Two DSM Data

In order to eliminate the calculation error of vertical variation caused by horizontal and height errors, the matching accuracy of both DSMs must be verified and evaluated before data processing. In this paper, visual recognition and coordinate comparisons are used, and landmark buildings are employed as checkpoints to verify the registration accuracy of the two DSMs and the Landsat 7 ETM+ images.

A total of 100 landmark buildings were selected as checkpoints (Figure 3). Then, each landmark building was checked to see if it was visually from different DSMs (AW3D30 DSM and SRTM DSM) to ensure that the location of the buildings in different DSMs was accurately matched. The result of the inspection was that the horizontal accuracy of the two datasets met the requirements. There was no need to perform a registration. This conclusion also provides convenience for others to follow up on this method.

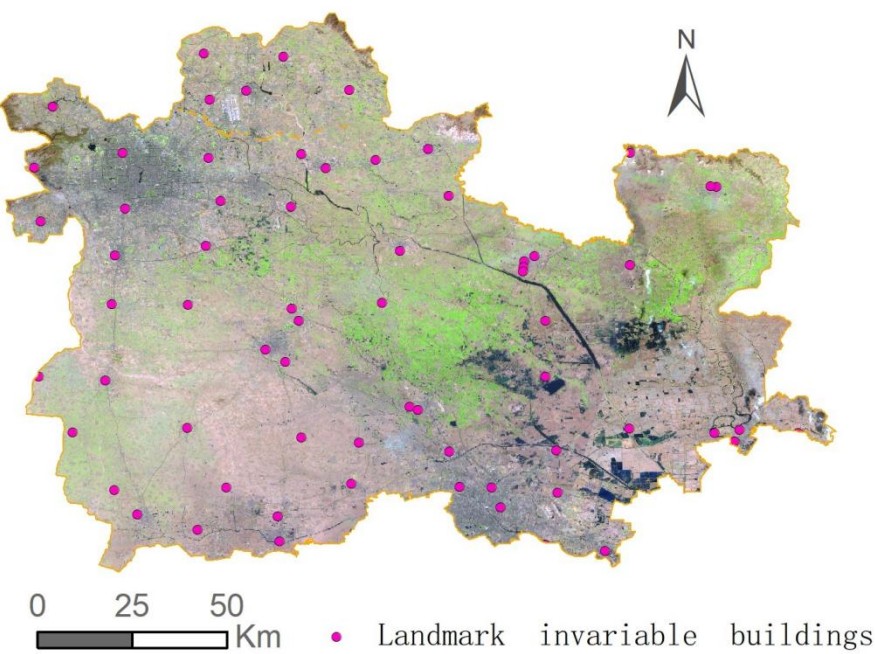

**Figure 3.** 100 landmark invariable buildings were selected as control points.

### 3.2. Detection Method of Height Change Information

After registration, through subtracting the historical elevation of the old DSM (SRTM DSM) from the new DSM (AW3D30 DSM), the grid pixel values of elevation change were obtained. The positive value indicates the area where the height increased, and the negative value indicates where the height decreased. As far as the change of urban terrain surface is concerned, the positive difference may mean an increase in ground heights caused by buildings. The negative difference may indicate excavation or demolished areas.

As shown in Figure 4, elevation changes are mostly within plus or minus 20 m. Moreover, the changes greater than 20 m are mainly concentrated in Beijing within the Second Ring Road and the Fifth Ring Road of Beijing and Tianjin Central City, indicating that the expansion of high-rise buildings in these places is rapid. Based on empirical judgment, the change of 20 m usually represents the construction of high-rise buildings (buildings with more than 6 floors), while the construction of urban areas is generally high-rise buildings with more than 6 floors. Therefore, in order to highlight the variation of construction factors and distinguish the variation of high-rise buildings from the height variation caused by non-construction factors, we assumed and manually classified them into −107−−100 (super large demolition), −100−−20 (large demolition), −20–0 (reduction of non-construction factors), 0–20 (increase of non-construction factors), and 20–181 (increase of construction factors). This is only the initial display to highlight the variation of architectural factors. The follow-up scientific evaluation is still behind.

### 3.3. Verification and Evaluation of Elevation Change Accuracy

According to the principle of statistics, the error of elevation difference ($\sigma_c$) is generally expressed by:

$$\sigma_c = \sqrt{\sigma_1^2 + \sigma_2^2},\tag{1}$$

where $\sigma_1$ and $\sigma_2$ are the errors of the first DSM dataset and the second DSM dataset, respectively. According to the previous study [52,53,58], for most of the two datasets, the probability density function is well characterized by a Gaussian distribution. It is known that the relative height errors of SRTM DSM and AW3D30 DSM in the study area are 4 m and 5 m, respectively. Therefore, based on the principle of Statistics (Formula (1)), the joint error of the height difference between the two DSMs is 6.4 m.

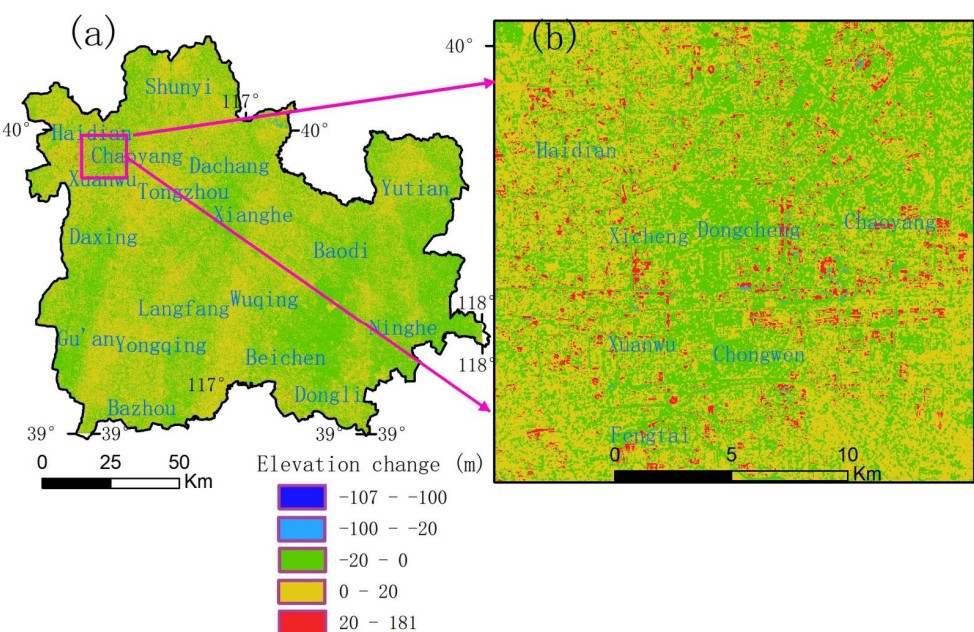

**Figure 4.** (**a**) Elevation change (m) obtained from the DSMs in the study area; (**b**) Enlarged view of elevation change in central urban area of Beijing.

In order to verify the consistency between the elevation change of the two DSMs and the increase of the actual vertical height of the building, 200 randomly distributed checkpoints were selected manually, as shown in Figure 5. The observation items included (1) verifying whether there was some historical information about building demolition or construction during that period, and (2) whether the calculated elevation change was consistent with the height change of building demolition or construction. Generally, if the difference between the two was less than 6 m, was is considered that the two were consistent, and the conclusion of judgment was set as T (true); otherwise, was is F (false).

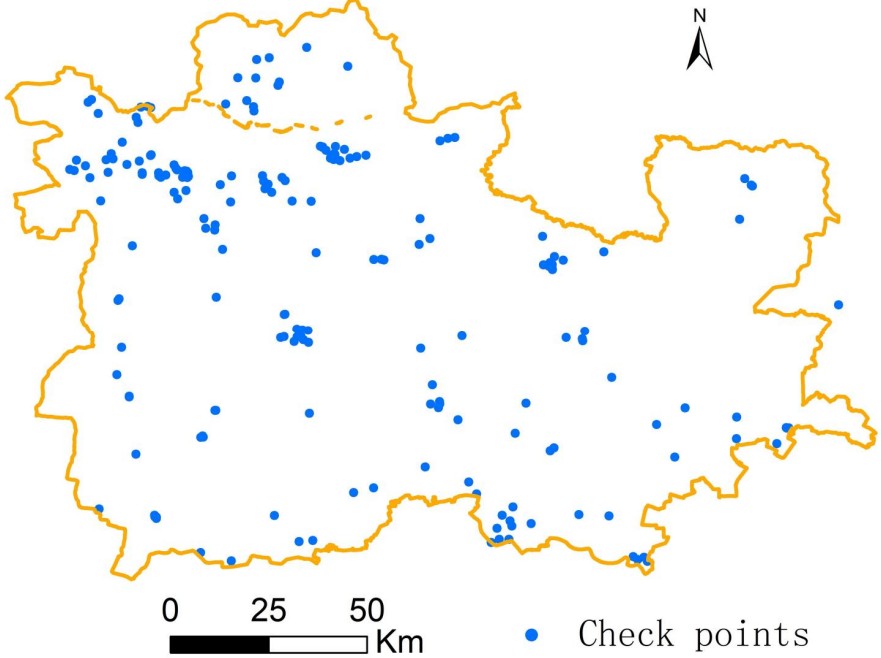

**Figure 5.** Distribution of verification points for the authenticity of elevation change.

Through building history retrieval and Google Map time-series images analysis, the authenticity of the corresponding relationship between the elevation changes and the

building expansion is verified accurately. Google Map time-series images analysis consists of several steps (Table 3). First, the point to be verified on Google map was located, then the changes of scenery and the height of building facilities on the historical images from 2000 to 2009 were compared, and combined with relevant construction information. Then, it was determined whether the difference between the actual height of building facilities and the elevation difference derived from DSMs was greater than 7 m, as well as whether it belonged to the true height change of buildings.

**Table 3.** Verification and evaluation of elevation change accuracy.

| Text Information | Height Change | Google Image before Change | Google Image after Change |
|---|---|---|---|
| Beijing Yintai Center Built in 2008 Increased by 141 m True | | | |
| Chengji building Build in 2007 Increased by 60 m True | | | |
| Tian an men No record of expansioin But increased by 17 m False | | | |
| Old building demolished Reduced 30 m in 2005 True | | | |
| Old building demolished Reduced 53 m in 2005 True | | | |

The verification result is shown in Figure 6. The point above 0 indicates where the height increased, and the size of its value indicates the amount of growth. The point below 0 indicates where the height decreased, and the size of its value indicates the amount of reduction. The verification result shows that the calculated elevation difference is inconsistent with the actual building height change in 7 random checkpoints, and the other 193 elevation changes are consistent with the building growth height. Although these tests are not enough to draw a conclusion on this consistency probability, some trends can be basically found, that is, among them, the probability of inconsistency in negative value

areas (areas with reduced height) is significantly greater than that in positive value areas (areas with increased height), and the smaller the negative value, the greater the error. This is essentially consistent with Ronald C. ESTOQUE's findings.

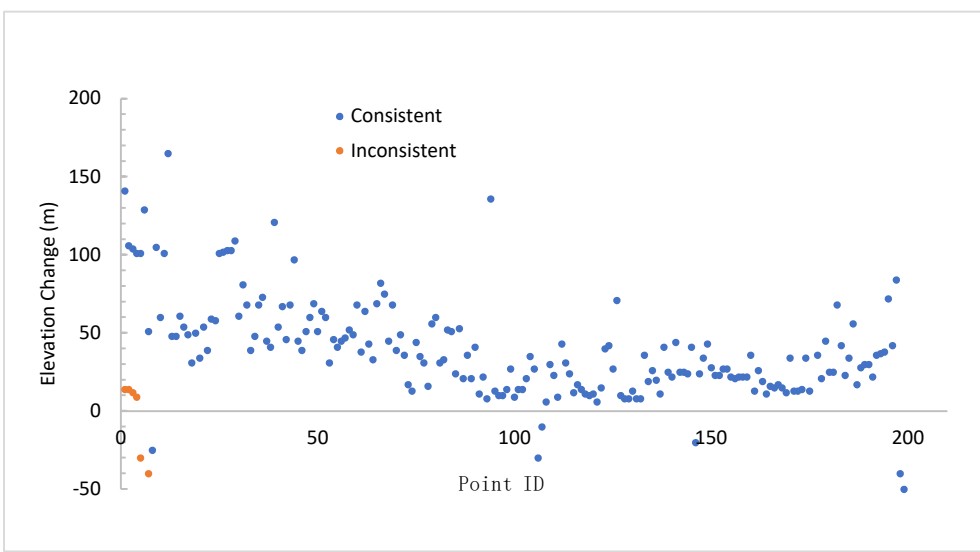

**Figure 6.** Authenticity probability of whether the height difference is consistent with the building expansion.

### 3.4. Exclusion Method of Non-Building Height Change

The exclusion of non-building height change is a crucial step. The objectivity of exclusion is directly related to the accuracy of building height change statistics. In this study, two methods were employed and compared to determine which was more reasonable. Firstly, the construction lands in two phases were extracted separately from Landsat 7 ETM+ images through the decision tree classification in ENVI software (ENVI 5.5 from ESRI, USA), which was mainly based on the different reflectivity of different ground objects in different electromagnetic bands. Then, the scopes of the construction lands at two phases were merged, which was regarded as the limited scope of the building height change statistics. Therefore, elevation changes outside this scope were considered non-building land changes, and therefore should be excluded. Secondly, in the second method, through the previous detection, in the plain area with a slope less than 5 degrees, the greater the height change, the greater the possibility of real building height change. Therefore, based on previous experience [52–59], elevation changes greater than a certain threshold were regarded as non-building changes, and therefore should be excluded.

Based on the first exclusion method, Figure 7b results from the elevation change retained only in the construction area, regarded as the building height change. The built-up area was obtained by merging the two-phase construction regions in April 2000 and April 2009, as shown in Figure 7a.

Based on the second exclusion method, non-building height changes were excluded by setting the exclusion threshold of elevation change, that is: (−4 m, 4 m), (−5 m, 5 m), (−6 m, 6 m), (−7 m, 7 m), (−8 m, 8 m), (−9 m, 9 m), (−10 m, 10 m), (−11 m, 11 m), (−12 m, 12 m), (−13 m, 13 m), (−14 m, 14 m), (−15 m, 15 m), (−16 m, 16 m), (−17 m, 17 m), (−18 m, 18 m), (−19 m, 19 m), and (−20 m, 20 m), respectively. For the purpose of saving space, only the results of (−7, 7) (Figure 8a), (−11, 11) (Figure 8b), (−14, 14) (Figure 8c), and (−20, 20) (Figure 8d) are shown in Figure 8. It can be seen that, with the increase of the exclusion threshold of the elevation change, the building height change pixels generally show a decreasing trend.

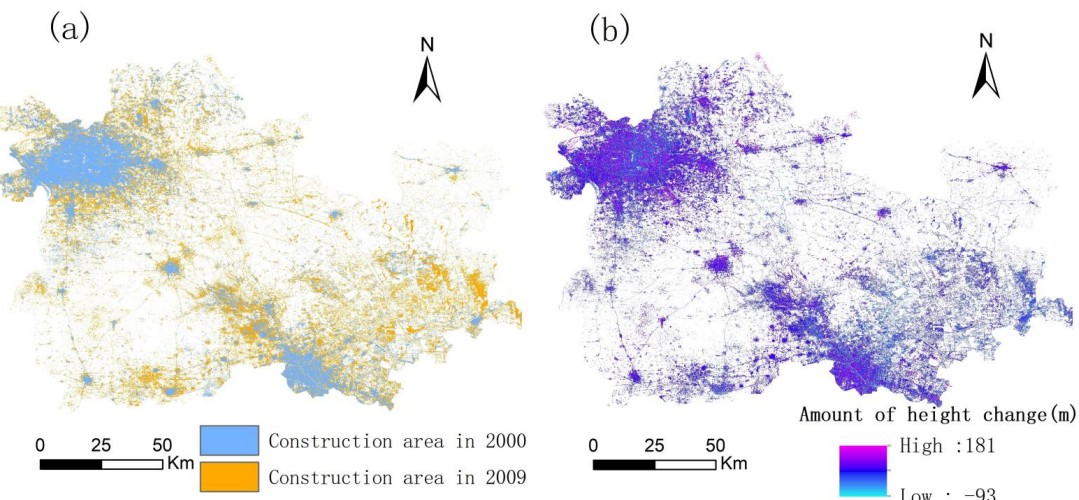

**Figure 7.** (**a**) The construction area extracted from the two original images. (**b**) The height change retaining only the construction area.

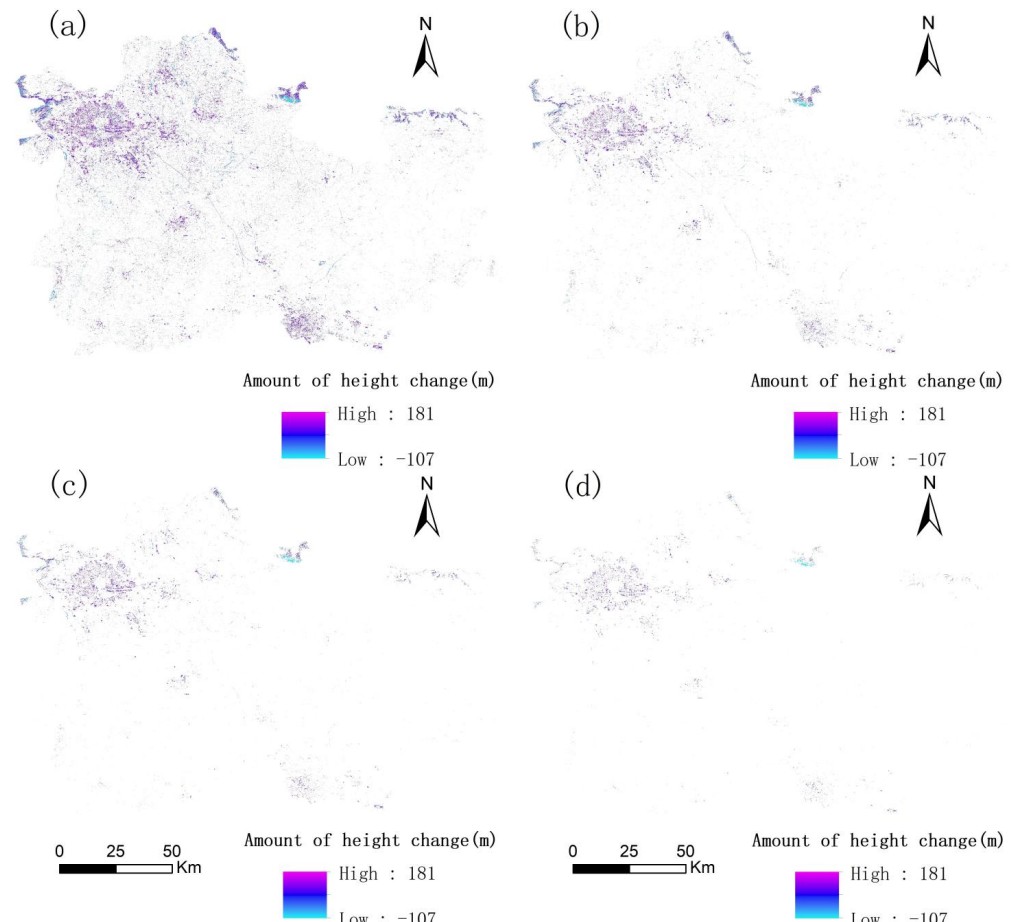

**Figure 8.** (**a**) Height changes of less than 7 m are excluded; (**b**) Height changes of less than 11 m are excluded; (**c**) Height changes of less than 14 m are excluded; (**d**) Height changes of less than 20 m are excluded.

After comparing these different results derived from the two methods, it can be seen that the basic distribution characteristics are almost the same, but the second method will gradually reduce the construction volume with the increase of the threshold of heigh change exclusion.

### 3.5. Calculation of Building Expansion or Demolition Amount

The methods of neighborhood block statistics and zoning statistics were used to highlight the differences between the total construction volume in different regions. Using block statistics, the input height variation raster was partitioned into non-overlapping blocks. The mean value within each block was assigned to all cells in the output block. Using zonal statistics, the mean value of the height variation raster within each county was obtained.

## 4. Results

### 4.1. Enhanced Display of Distribution Characteristics of 3D Building Expansion

Continuous neighborhood block statistics based on the result of Section 3.4 (here, only taking the results of Figure 8c as an example) were carried out for every 30 pixels, and the results are displayed in Figure 9. Such filter processing is suitable for visual observation and comparison of building expansion differences in different regions (independent of administrative divisions), although the range of elevation variation is compressed and reduced after filtering. The results in Figure 9 indicate that the areas with significant building expansion are primarily concentrated in the central regions of the big cities, such as the central urban area of Beijing and Tianjin.

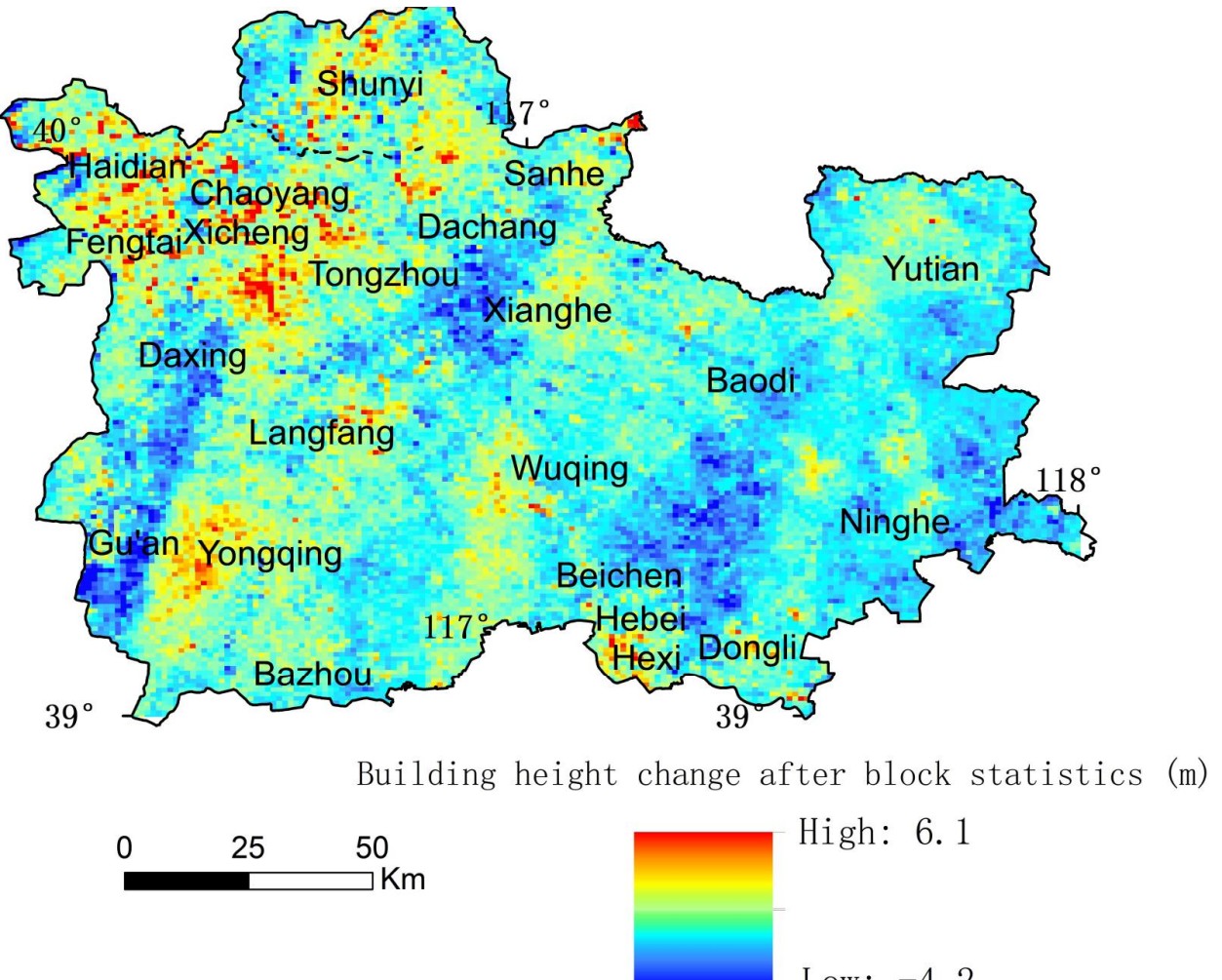

**Figure 9.** The result of building expansion by continuous block statistics.

*4.2. Zoning Statistics and Comparison of Three-Dimensional Building Expansion*

The zoning statistics of each county in Beijing–Tianjin–Hebei Plain were also carried out. The average building height expansion per unit area (ABHE) in each county, which is dimensionless, was calculated separately. Table 4 shows the statistical ABHE based on different height change exclusion methods.

**Table 4.** ABHE values of each county in Beijing based on different exclusion thresholds.

| Threshold \ County | Tongzhou | Daxing | Chaoyang | Haidian | Dongcheng | Xicheng | Xuanwu | Chongwen | Fengtai | Shijing | Shunyi |
|---|---|---|---|---|---|---|---|---|---|---|---|
| Non urban land | 0.590816 | 0.500022 | 2.073694 | 2.02177 | 1.837113 | 3.236168 | 2.837574 | 1.867613 | 1.99719 | 2.432005 | 0.506084 |
| 4 | 0.788478 | 0.520568 | 2.045749 | 1.950714 | 2.248053 | 3.412032 | 3.053046 | 1.972498 | 1.67909 | 1.981709 | 0.77858 |
| 5 | 0.612751 | 0.401682 | 1.892196 | 1.740174 | 2.39307 | 3.378006 | 3.079197 | 2.090235 | 1.479749 | 1.802721 | 0.610155 |
| 6 | 0.468008 | 0.313474 | 1.750427 | 1.558582 | 2.504611 | 3.303604 | 3.066669 | 2.190896 | 1.311428 | 1.645083 | 0.471058 |
| 7 | 0.366101 | 0.256763 | 1.63014 | 1.417425 | 2.573855 | 3.230102 | 3.013584 | 2.245899 | 1.177988 | 1.497451 | 0.369053 |
| 8 | 0.296063 | 0.218761 | 1.52952 | 1.306598 | 2.598711 | 3.135227 | 2.958503 | 2.251876 | 1.071545 | 1.371032 | 0.299834 |
| 9 | 0.248759 | 0.19184 | 1.444442 | 1.215632 | 2.591838 | 3.041395 | 2.902756 | 2.229228 | 0.984002 | 1.259452 | 0.252396 |
| 10 | 0.215283 | 0.169357 | 1.368507 | 1.138959 | 2.564519 | 2.948203 | 2.833699 | 2.182717 | 0.91575 | 1.170472 | 0.219662 |
| 11 | 0.187433 | 0.151088 | 1.303423 | 1.072297 | 2.532159 | 2.861242 | 2.766364 | 2.11576 | 0.856773 | 1.093623 | 0.192865 |
| 12 | 0.164982 | 0.134399 | 1.246214 | 1.00917 | 2.474513 | 2.77549 | 2.68153 | 2.030737 | 0.808022 | 1.02087 | 0.171771 |
| 13 | 0.145556 | 0.119036 | 1.19295 | 0.952124 | 2.433963 | 2.682795 | 2.592742 | 1.964949 | 0.762726 | 0.952709 | 0.154354 |
| 14 | 0.13006 | 0.105553 | 1.14324 | 0.899244 | 2.374026 | 2.598085 | 2.514876 | 1.880241 | 0.719627 | 0.888 | 0.138516 |
| 15 | 0.116265 | 0.094477 | 1.097886 | 0.848527 | 2.307875 | 2.515153 | 2.435954 | 1.791534 | 0.682002 | 0.821792 | 0.123545 |
| 16 | 0.104607 | 0.084606 | 1.053467 | 0.799435 | 2.247738 | 2.425586 | 2.350219 | 1.716712 | 0.647664 | 0.763449 | 0.110542 |
| 17 | 0.093981 | 0.075738 | 1.010565 | 0.753673 | 2.192755 | 2.345592 | 2.275681 | 1.626837 | 0.613655 | 0.705465 | 0.098463 |
| 18 | 0.08435 | 0.068232 | 0.971502 | 0.705975 | 2.121678 | 2.26664 | 2.196484 | 1.548915 | 0.582381 | 0.657111 | 0.08855 |
| 19 | 0.077406 | 0.061812 | 0.93518 | 0.662075 | 2.066008 | 2.184323 | 2.102059 | 1.475307 | 0.553532 | 0.606684 | 0.079863 |
| 20 | 0.071012 | 0.0559 | 0.900591 | 0.622577 | 1.997451 | 2.105087 | 2.006851 | 1.383948 | 0.526469 | 0.563548 | 0.072142 |

Table 4 and Figure 10 show that these results are roughly consistent. They show a common trend. In terms of the ABHE of each county, the overall trend reflected by these statistical algorithms after excluding non-building height changes is roughly the same: Chaoyang, Dongcheng, Xicheng, Xuanwu, Chongwen, Nankai, Heping, and Hexi have a significant speed of construction expansion volume. In contrast, the average expansion speed of Gu'an, Ninghe, and Baodi counties is the slowest, even almost negative. The imbalance of the average expansion is surprisingly consistent with the current urban development stage and the characteristics of each county: Chaoyang, Haidian, Dongcheng, Xicheng, Xuanwu, Chongwen, Nankai, Heping, and Hexi are all highly developed and first-tier cities; however, Gu'an, Baodi, and other places are the third or fourth-tier cities, with relatively slow development, and mainly low-rise buildings and bungalows, rarely more than twenty stories.

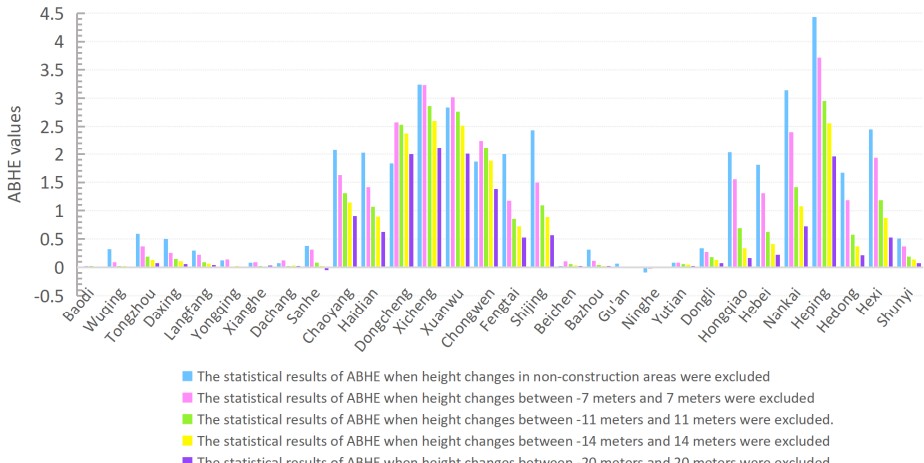

**Figure 10.** Comparison of building expansion between counties of Beijing–Tianjin–Hebei.

The differences between these statistical results are: (1) the blue bars are always high, while the purple bars are always low, while the pink, green, and yellow bars are relatively moderate; (2) this trend is most obvious in Fengtai and Shijing districts. However, in Dongcheng District and Chongwen District, this trend has been broken, and the statistical results of the blue bars are lower than other bars.

## 5. Discussion

### 5.1. Discussion on the Exclusion of Non-Building Height Variation

The elimination of elevation changes of non-building factors has always been the bottleneck. At present, remote sensing images are used to extract building land areas, and only the height change of the building area is calculated in the statistics. However, due to the limited accuracy of extracting construction land from images, some non-building height change factors will inevitably interfere with the calculation results. On the other hand, setting the elevation change threshold to exclude the non-building height change makes it difficult to exclude a sizeable non-building height change in particular areas. Therefore, combining the two methods to obtain more objective results is better.

### 5.2. Comparison with Other Author's Research

Xuecao Li presented a horizontal urban land dynamic of Beijing City between 1984–2013 based on image classification and change detection [20]. Shisong Cao derived the horizontal urban expansion in Beijing–Tianjin–Tangshan metropolitan region from 1990 to 2015, based on image classification [22]. Both of them have certain reference significance, but they involve horizontal dynamic monitoring, and do not involve vertical dynamic changes.

In order to compare the difference between horizontal expansion and vertical expansion in Beijing Tianjin Hebei region, the construction areas in the two phases were extracted through image classification. Then, the horizontal construction expansion volume of each county was calculated through regional building area statistics, as shown in Figure 11. It is significantly different from the vertical expansion statistical result in this paper (Figure 10). The areas with the largest expansion are Beichen, Ninghe, Dongli, Bazhou, Wuqing, etc. One possible reason may be related to the stage of regional development. Beichen, Ninghe, Dongli, Bazhou, Wuqing, and others belong to the fourth-tier counties, with relatively slow growth. Their undeveloped areas are vast, and their expansion is mainly horizontal. On the contrary, there is no space for horizontal expansion in the central urban area, but primarily vertical expansion. Obviously, the real three-dimension expansion speed is difficult to reveal only through the horizontal expansion statistics interpreted by the images.

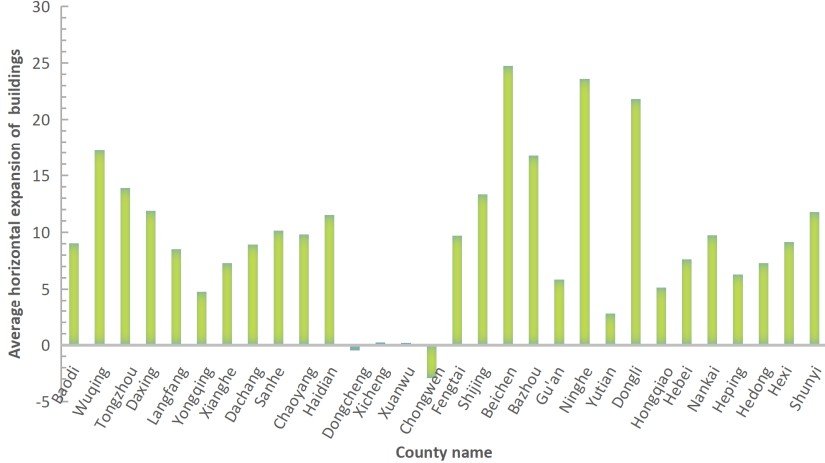

**Figure 11.** Comparison of building horizontal expansion between counties. Note: The average horizontal expansion of buildings in Figure 11 expresses the expansion ratio of construction land per unit area, which is dimensionless.

*5.3. Comparison with the Result Derived from Statistical Yearbook*

In order to verify the rationality of this method, the average completed building floor space per unit area (ACBFS) in each county derived from the Beijing Statistical Yearbook (BSY) [41–50] of all counties in Beijing between 2000 and 2009 was employed as an inspection standard for comparison with the average building height expansion per unit area (ABHE) in each county of Beijing. Since it is difficult to obtain the ACBFS data of Hebei and Tianjin, the following analyses only take some counties in Beijing as examples. Table 5 shows the real ACBFS in each county of Beijing, which is dimensionless.

**Table 5.** The values of real ACBFS in each county of Beijing.

|  | Tongzhou | Daxing | Chaoyang | Haidian | Dongcheng | Xicheng | Xuanwu | Chongwen | Fengtai | Shijing | Shunyi |
|---|---|---|---|---|---|---|---|---|---|---|---|
| Real ACBFS | 2.52 | 2.20 | 22.57 | 11.74 | 40.96 | 45.13 | 52.17 | 46.10 | 12.69 | 11.96 | 1.75 |

Figure 12 is the scatter plot of statistical ABHE versus real ACBFS. It shows the variation trend of the coefficient of determination ($R^2$). With the increase in the elevation change exclusion threshold, the $R^2$ value gradually increases until it reaches the peak value of 0.9436, when the exclusion threshold goes to (−14 m, 14 m). After that, it began to decline gradually. It proves that 0.9436 is the best correlation coefficient value. Accordingly, the second exclusion method in Section 3.4 with a 14-m threshold is proved to be more objective in this study area.

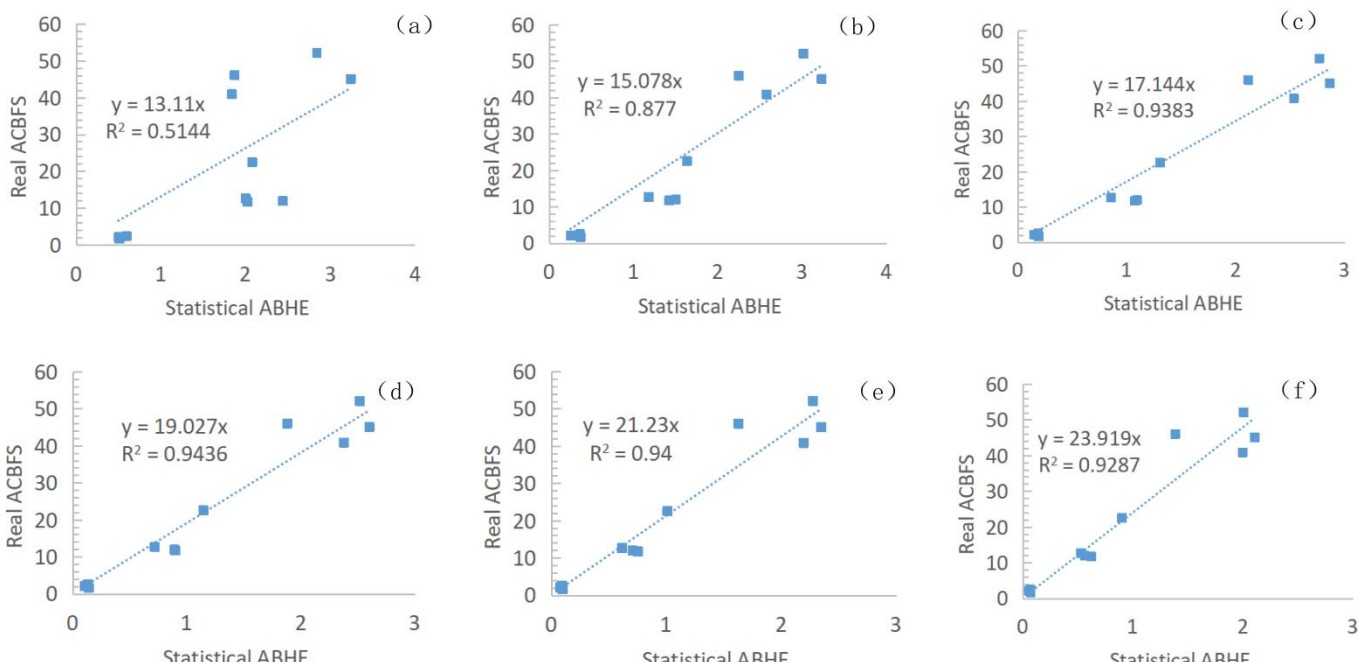

**Figure 12.** The fit curve between statistical results and real situations. (**a**) Fitting analysis result of the Statistical ABHE (based on the first non-building exclusion method) and the real ACBFS; (**b**) Fitting analysis result of the Statistical ABHE (based on the second non-building exclusion method and the exclusion threshold is 7 m) and the real ACBFS; (**c**) Fitting analysis result of the Statistical ABHE (based on the second non-building exclusion method and the exclusion threshold is 11 m) and the real ACBFS; (**d**) Fitting analysis result of the Statistical ABHE (based on the second non-building exclusion method and the exclusion threshold is 14 m) and the real ACBFS; (**e**) Fitting analysis result of the Statistical ABHE (based on the second non-building exclusion method and the exclusion threshold is 17 m) and the real ACBFS; (**f**) Fitting analysis result of the Statistical ABHE (based on the second non-building exclusion method and the exclusion threshold is 20 m) and the real ACBFS.

## 6. Conclusions

In this paper, the study of urban expansion reveals that the areas with the rapid centralized growth of construction volume during 2000–2009 mainly occurred between the Second Ring Road and the Fifth Ring Road of Beijing, followed by Yizhuang, Shunyi, Tianjin Central City, Langfang, etc. At the same time, there is also a significant imbalance in the amount of average building expansion among counties of the Beijing–Tianjin–Hebei plain. Zonal statistics of each county show that Chaoyang, Dongcheng, Xicheng, Xuanwu, Chongwen, Nankai, Heping, and Hexi have a significant construction expansion volume. However, other counties present a relatively slow rate of building expansion. The difference in building expansion volume correlates with each county's actual scale, urban development stage, and policy at that period.

This method has been tested by the data of the Beijing Statistical Yearbook, and the results are consistent, which proves that this method is reliable to a certain extent. At present, the actual statistics of construction expansion of counties in Hebei and Tianjin Statistical Yearbooks are missing. Our research can make up for this deletion. At the same time, this application can also be extended to any other plain urban area in the world. However, it is not suitable for mountainous cities due to the limitation of current data accuracy. With the continuous improvement of DSM accuracy and the increasing enrichment of DSM time series, this method will play a more powerful role in urban building height change detection and global urban expansion in metropolitan areas.

**Author Contributions:** Conceptualization, Y.W. and N.Y.; methodology, Y.W. and P.D.; software, Y.W. and S.L.; validation, Y.W. and D.Z.; formal analysis, Y.W.; investigation, Y.W.; resources, Y.W.; data curation, Y.Z.; writing—original draft preparation, Y.W.; writing—review and editing, Y.W. and P.D.; visualization, Y.W.; funding acquisition, Y.W. All authors have read and agreed to the published version of the manuscript.

**Funding:** This research was funded by the Science and Technology Research Project Fund of Colleges and Universities in Hebei Province (Grant No. ZC2021210), the Fundamental Research Funds for the Central Universities (Grant No. ZY20215153 and ZY20180101).

**Institutional Review Board Statement:** Not applicable.

**Data Availability Statement:** The data produced by this work are freely available in the Science Data Bank at https://www.scidb.cn/en/anonymous/VVZOVnp1 (accessed on 19 February 2022).

**Acknowledgments:** The authors would like to thank the following colleagues: Dean B. Gesch and C.-T. Yan, for their valuable suggestions in this study; Wang Cheng and his students for their data support; Thank Yi Zhao and Keqi Zhang for the collation work of chart data in this study. Thanks are also due to the European Space Agency (ESA) for their data sharing and explanation of data quality.

**Conflicts of Interest:** The authors declare no conflict of interest.

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
