# Peer review of "Urban Expansion Monitoring Based on the Digital Surface Model—A Case Study of the Beijing–Tianjin–Hebei Plain"

_applsci, doi:10.3390/app12115312_

Round 1
Reviewer 1 Report
Dear authors, find below my comments to your MS:
Two “ --” in the title
Introduction overall seems short for a research paper. I believe that there is context and valuable information that could be added about the topic itself and the importance of the research
L37 – Middle of the sentence with Upper letter.
L44 – What is an advanced stage of a big city?
L44/45 - Lack of reference
L52 – “.” before the reference
L52/53 – What more details? Instead of another example (previously found in the introduction text), I believe a “such as” with additional references to other details that need to be studied would be interesting
The study area map is not readable. Its quality needs to be improved.
Study area details should be added (Population; urbanization rate; land use…). Again this section is way too short and lacks content
Data – Authors indeed described the characteristics of the data sources, but they do not detail the data used as inputs for their research as much. There is no reference to other data that the reader finds in figure 2.
Which software(s) were used for this research?
L92-95 – Different descriptions are given in the 5 major research steps on L92-95 and afterwards on each subchapter
Methodology lacks detail about how the authors conducted their research. Example – There are no shared details about how the image classification or change detection was performed. Or Google Map time-series images analysis is never referred to. Therefore, methodological doubts may arise. This section seems more informative rather than descriptive
4.1 section is not a result, but a method used for accuracy
Figure 4 - Which classifications method did the authors use? If manual, it does not suit the purpose as there is not any scientific value. See for instance that the largest category falls out compared with the other categories´ interval range.
L159 -161 – This sentence is not well structured
L161 – Double space
Table 2 – This table is irrelevant for the core presentation of the paper.
The same applies to figure 7 which is input data and therefore not interested as it does not provide any outcome
This paper has too many figures, and some of them are irrelevant. Authors will need to find a strategy to clean the MS presentation to be easier for the reader to follow your findings
I can´t see the purpose for adding table 4, Figures 12 and 14 in the discussion section. It is unappropriated to add those figures into this section
In the discussion section, the results of the MS are not compared with other author’s research
Where is figure 13?
Author Response
Dear reviewer:
We appreciate for putting forward so many specific modification suggestions to this article. That will be a great help to the improvement of this paper.
We will add our replies to your comments item by item. At the same time, the modified contents have been added in applsci-1664370, and emphasized in red font.
We would like to thank you very much for your review of this manuscript.
With best wishes,
Yours sincerely,
Yanping Wang
Department of Ecology and Environment
Institute of Disaster prevention (CIDP)

Reviewer 2 Report
Thanks a lot for your manuscript submission to MDPI Applied Sciences. After careful review, my comprehensive evaluation on this paper is a very good set of work. The study is fair, complete and convincing, the literal writing is also moderately acceptable. Hence, this research article, can be recommended as "Acceptance with Minor Edits".

Author Response
Dear reviewer:
Appreciate a lot for your review and your acceptance. We will have more confidence to do the next step with your support and encouragement.
We would like to thank you very much for your review of this manuscript.
With best wishes,
Yours sincerely,
Yanping Wang
Department of Ecology and Environment
Institute of Disaster prevention (CIDP)
Reviewer 3 Report
Dear Authors,
I have reviewed the paper entitled "Urban expansion monitoring based on digital ground model -- a case study of Beijing Tianjin Hebei Plain". The manuscript presents a study of Urban Expansion based on DEM and Satellite Imagery. I think the paper needs few improvements before potential publication.
- Line 34. What are this statistics ? There are many references about statistics in the text.
- The methodology is about calculating buildings height ?
- How were the coordinates compare in accuracy assessment ?
- subchapter 3.3 - what about normal distribution ?
- How were ground control points measure ?
- Table 2 is blurred and hard to read.
- Fig. 6 What does the point mean below 0 level ? There is only one.
- I think the Authors should add more information about the methodology itself (image processing, statistics evaluation, normal distribution (perhaps). For me it is not very clear, whether the methodology is touching the brand of science.
Author Response

(The authors gave the same response as above.)

Reviewer 4 Report
The article addresses a highly relevant issue. Urbanisation processes are problematic in many regions of the world. Therefore, more recent international literature is needed. The basics of the topic should be more developed. References and a broader perspective should be made on similar agglomerations (in Europe, America or Africa).
The conclusion should be extended: to the novelty and limitations of the research.
Emphasis should be placed on the results that could be relevant elsewhere.
Author Response
Dear reviewer:
Appreciate that you give us valuable suggestions. We re-studied and supplemented more literature in the introduction section and conclusion section.
We will add our replies to your comments point by point. At the same time, the modified contents have been added in applsci-1664370, and emphasized in red font.
We would like to thank you very much for your review of this manuscript.
With best wishes,
Yours sincerely,
Yanping Wang
Department of Ecology and Environment
Institute of Disaster prevention (CIDP)

Round 2
Reviewer 1 Report
Dear authors,
You have improved your MS and tackled the issues that I´ve pointed out in the 1st review. Moreover, you provided a clear and detailed explanation for each comment.
Congratulations on the effort.
Still, the study area map has low cartographic quality. Check the scale that overlays the inset map and the coordinates; the labels are difficult to see where cities are located as for instance there is not a reference point for each label; the upper map overlays the coordinates; the legend is unplaced and the map as a product does not look a simple and good quality figure. The figure isn´t appropriate to be inserted in a scientific article. Please review it carefully.
Also, confirm that the Figure 4 answer is well explained in the text as it is in the comment.
Author Response
Thank you very much for your reminder. After careful consideration, we now remended the study area map. Avoided the overlaps, stressed the labels, added a reference point layer for counties, deleted the upper map, and adjusted the legend.
Thank you very much for your reminder. Figure 4 has been well explained in the text (section 3.2).
Reviewer 3 Report
Dear Authors,
Thank You for revised version of Your manucript. I see that You put a lot of effort to improve it.
In my opinion a key to obtain a full spectrum of research is a proper classification. "Initial classification" is a very general and the meaning of it should be emphasized. That's the only comment I have.
Author Response
Thank you very much for your comment. After careful consideration, we deleted the text "classification." So now it changed from the "initial classification display" to "initial display." Actually, we only displayed the results of the elevation changes at this step.